# Relationship between Sleep and Hypertension: Findings from the NHANES (2007–2014)

**DOI:** 10.3390/ijerph18157867

**Published:** 2021-07-25

**Authors:** Chunnan Li, Shaomei Shang

**Affiliations:** School of Nursing, Peking University, 38 Xueyuan Road, Hai Dian District, Beijing 100191, China; li_chunnan@bjmu.edu.cn

**Keywords:** sleep duration, sleep pattern, hypertension, epidemiology

## Abstract

Background: To evaluate the association of sleep factors (sleep duration, self-reported trouble sleeping, diagnosed sleep disorder) and combined sleep behaviors with the risk of hypertension. Methods: We analyzed 12,166 adults aged 30–79 years who participated in the 2007–2014 National Health and Nutrition Examination Survey. Sleep duration, self-reported trouble sleeping and sleep disorders were collected using a standardized questionnaire. We included three sleep factors (sleep duration, self-reported trouble sleeping and sleep disorder) to generate an overall sleep score, ranging from 0 to 3. We then defined the sleep pattern as “healthy sleep pattern” (overall sleep score = 3), “intermediate sleep pattern” (overall sleep score = 2), and “poor sleep pattern” (0 ≤ overall sleep score ≤ 1) based on the overall sleep score. The definition of hypertension was based on self-reported antihypertensive medication use or biological measurement (systolic blood pressure ≥140 mm Hg and/or diastolic blood pressure ≥90 mm Hg). We used weighted logistic regression models to investigate the associations between sleep and hypertension. Results: The overall prevalence of hypertension was 37.8%. A short sleep duration (OR = 1.20, 95% CI: 1.08 to 1.33, *p* = 0.001), self-reported trouble sleeping (OR = 1.45, 95% CI: 1.28 to 1.65, *p* < 0.001) and sleep disorder (OR = 1.33, 95% CI: 1.07 to 1.66, *p* = 0.012) were related to the risk of hypertension. Poor sleep patterns were closely correlated with the risk of hypertension (OR = 1.90, 95% CI: 1.62 to 2.24). Conclusions: Participants with poor sleep patterns were associated with an increased risk for hypertension.

## 1. Introduction

Hypertension is the key preventable risk factor for cardiovascular disease (CVD) and all-cause mortality worldwide, and rates of hypertension are increasing due to population aging and unfavorable lifestyle factors [1].

In addition to the main factors related to diet and physical activity, the association between sleep and hypertension has been recently studied extensively. Owing to high interindividual and intraindividual variations in sleep at different ages across the lifespan [2], the American National Sleep Foundation has recommended a 7–9 h sleep duration for adults [3]. Meta-analyses have found that short and long sleep durations are associated with a higher risk of hypertension [4,5]. A cross-sectional study with over 700,000 people found that both short and long sleep durations were associated with increased hypertension risk [6]. Short sleep duration, defined as less than 7, 6 or 5 h of sleep per night, is associated with the risk of hypertension [7]. A longitudinal study based on a Chinese population found that a short sleep duration, but not a long sleep duration, was a risk factor for hypertension [8]. Conversely, some studies have linked a long sleep duration to hypertension [9]. Still others have shown that there is no correlation between a long sleep duration and hypertension [10]. Despite inconsistencies, these findings suggest that sleep duration may play a role in the risk of hypertension. In addition to the length of sleep, the association between sleep complaints, sleep disorders and hypertension has received increasing attention in recent years. Two large cohort studies demonstrated that obstructive sleep apnea (OSA) was associated with an increased risk of incident hypertension [11,12]. In the Wisconsin Sleep Cohort Study, participants with severe OSA (apnea–hypopnea index > 15/h) had a 3.2-fold increase in the odds of developing hypertension [11]. Long-term continuous airway positive pressure therapy is associated with a reduced risk of hypertension [12].

Sleep behaviors are multifaceted, and individual sleep behaviors are usually related to each other in a compensatory way [13,14]. In most previous studies, sleep behaviors were considered individually, without considering the complexity and correlation between sleep parameters [13]. Limited studies have been performed to assess sleep behaviors jointly; these have suggested that sleep duration in combination with other unhealthy sleep behaviors, such as trouble sleeping and sleep disorders, is associated with a higher risk of CVD and hypertension [15,16,17].

Based on the nationally representative survey, we assessed the association between each sleep trait (sleep duration, troubled sleeping and sleep disorder) and the risk of hypertension. We also estimated the association between joint sleep factors (defined as a “sleep pattern”) and hypertension.

## 2. Materials and Methods

### 2.1. Study Population

The National Health and Nutrition Examination Survey (NHANES) is a nationally representative cross-sectional survey conducted in the United States and is based on a stratified multistage random sampling design. This study includes data from the NHANES 2007–2008, 2009–2010, 2011–2012 and 2013–2014 cycles.

After excluding respondents with CVD (including congestive heart failure, heart failure, coronary heart disease, angina pectoris, heart attack or stroke) and diabetes (yes/borderline), 12,166 participants aged 30–79 years with complete and reliable information (demographics, health behaviors, body measurements and disease information) were included.

### 2.2. Measurement

#### Definition of Hypertension

Participants were considered to have hypertension for any of the following reasons: if they responded “yes” to the question “Have you ever been told by a doctor or other health professional that you have hypertension, also called high blood pressure?”; if they self-reported antihypertensive drug use; or if they had a high biological measurement value (systolic blood pressure ≥140 mm Hg and/or diastolic blood pressure ≥90 mm Hg) [18]. Three consecutive blood pressure readings were obtained, and an additional determination may have been measured if a blood pressure measurement was interrupted or incomplete. In our study, we used the mean of the readings to define hypertension.

### 2.3. Assessment of Sleep Factors and Definition of a Sleep Pattern

Nighttime sleep hours were obtained by the response to “How much sleep do you usually get at night on weekdays or workdays?”. The sleep duration was categorized as short (<7 h per night), normal (7–9 h per night), or long (>9 h per night), and 7–9 h per night was used as the reference group. The response to “Have you ever told a doctor or other health professional that you have trouble sleeping?” was used to assess trouble sleeping. The response to “Have you ever been told by a doctor or other health professional that you have a sleep disorder?” was used to assess sleep disorder. Sleep factors (sleep duration, self-reported trouble sleeping and diagnosed sleep disorder) were included to compute an overall sleep score. Low-risk sleep was defined as sleeping 7–9 h per night without trouble sleeping or sleep disorder. For each sleep factor, low risk was assigned as 1, and high risk was assigned as 0. Finally, we obtained a total score ranging from 0 to 3 and defined the overall sleep pattern as “poor sleep pattern” (0 ≤ overall sleep score ≤ 1), “intermediate sleep pattern” (overall sleep score = 2), and “healthy sleep pattern” (overall sleep score = 3).

### 2.4. Other Covariates

A structured questionnaire was used to collect the following sociodemographic information: gender (male, female), age (30–44, 45–59, 60–79 years), education level (less than 9th grade, 9–11th grade, 12th grade with no diploma, high school graduate/GED or equivalent, some college or AA degree, college graduate or above), marital status (married/living with partner, widowed, divorced, separated, never married), covered by health insurance (yes, no), race (Mexican American, other Hispanic, non-Hispanic Black, other race—including multi-racial, non-Hispanic White). BMI was calculated as weight (kg) divided by height squared (kg/m^2^). The physical activity condition was based on the Global Physical Activity Questionnaire (GPAQ), which includes questions related to daily activities, leisure time activities, and sedentary activities. The metabolic equivalent fraction was calculated according to suggested metabolic equivalent (MET) scores for the activities (available at: https://wwwn.cdc.gov/Nchs/Nhanes/2013–2014/PAQ_H.htm, accessed on 1 May 2021). Sedentary time refers to the time spent sitting in a typical day, excluding sleep. Dietary intake was determined using valid 24-hour dietary recalls, as detailed in a previous study [19]. Mellen’s DASH diet score was established based on 9 target nutrients (total fat, saturated fat, protein, fiber, cholesterol, calcium, magnesium, sodium and potassium), and the scoring methods were built based on previous studies [20,21]. Daily alcohol consumption was divided into 0–24.9 and 25.1–550.1 gm. Smoking status was obtained from the question “Have you smoked at least 100 cigarettes in your entire life?” (yes/no).

### 2.5. Statistical Analysis

The baseline characteristics of the participants were categorized by sleep patterns and hypertension status.

We applied the survey weight from the mobile examination center (MEC exam weight) to all analyses to account for the cluster sample design, oversampling, poststratification, survey nonresponse and sampling frame, as suggested. The weight of the survey allows it to be extended to the civilian noninstitutionalized US population [22,23]. Weighted logistic regression was used to calculate the odds ratios (ORs) and 95% confidence intervals (95% CIs) to assess the relationship between each sleep factor and hypertension. The same analyses were performed to explore the relationship between sleep patterns and hypertension.

Statistical analyses were performed with STATA version 14.0 (Stata Corp LP, College Station, TX, USA). The forest graphs were plotted using R version 3.5.3. A *p*-value < 0.05 was defined as statistically significant.

## 3. Results

### 3.1. Baseline Characteristics of Participants

The baseline characteristics of the study population according to sleep patterns are listed in Table 1. Of the 12,166 participants (47.3% males and 52.7% females, mean (SD) age, 49.9 [13.2] years), 46.6%, 37.3% and 13.8% had a healthy, intermediate and poor sleep pattern, respectively (Table 1). Participants with a poor sleep pattern had a higher BMI and were more likely to be female, middle aged (45–59), and living alone, and engaged in less physical activity and more sedentary time, and were more likely to be heavy smokers (Table 1). The overall prevalence of hypertension was 37.8% (38.3% in males, 37.3% in females), and participants with poor sleep patterns appeared to have a higher prevalence of hypertension (Table 2).

### 3.2. Associations of Sleep with Risk of Hypertension

Figure 1 depicts the association between each sleep factor and hypertension, including the adjustments. In the age- and gender-adjusted model, a short sleep duration (OR = 1.28, 95% CI: 1.15 to 1.41, *p* < 0.001), long sleep duration (OR = 1.40, 95% CI: 1.04 to 1.89, *p* = 0.026), self-reported trouble sleeping (OR = 1.46, 95% CI: 1.29 to 1.64, *p* < 0.001) and sleep disorder (OR = 1.46, 95% CI: 1.18 to 1.82, *p* = 0.001) were all associated with the risk of hypertension. After additionally adjusting for race, marital status, education level, health insurance, BMI, physical activity, sedentary time, smoking status, alcohol intake, and the DASH index (fully adjusted models), these associations remained statistically significant except for a long sleep duration (OR = 1.22, 95% CI: 0.89 to 1.66, *p* = 0.215).

Figure 2 shows the joint effect of three sleep factors (sleep duration, self-reported trouble sleeping and sleep disorder). There was an increasing trend between a worse sleep pattern and the risk of hypertension. In the partially and fully adjusted models, compared to those with a healthy sleep pattern, participants with a poor sleep pattern always had the highest risk of hypertension (OR = 1.90, 95% CI: 1.62 to 2.24, *p* < 0.001).

### 3.3. Associations of Sleep with Risk of Hypertension after Age Stratification

We further explored the association between sleep patterns and hypertension after stratifying by age (30–44, 45–59, 60–79 years) (Figure 3). When these three sleep factors were considered jointly by establishing the participants’ sleep patterns, there was a substantial positive correlation between poor sleep patterns and the risk of hypertension regardless of age.

## 4. Discussion

In the nationally representative survey, we observed that sleep factors—short sleep duration, self-reported trouble sleeping and sleep disorder—were each associated with an increased risk of hypertension. In addition, a positive dose-response relationship was identified between poor sleep pattern and a higher risk of hypertension.

Our findings are in agreement with those of previous studies. Short sleep duration was confirmed to be a risk factor for the incidence of hypertension [24,25], and the risk of hypertension was reduced by 0.3207% for every 1 h of lengthening the sleep time [26]. In addition, longitudinal analyses of the first NHANES survey (*n* = 4810) indicated that a short sleep duration led to an increased risk of developing or dying of hypertension [27]. A systematic review of 13 cross-sectional and longitudinal studies with a combination of 225,858 participants found a significant association between a short sleep duration and hypertension (RR, 1.23; 95% CI, 1.06 to 1.42; *p* = 0.005) but a nonsignificant relationship between a long sleep duration and hypertension (RR, 1.02; 95% CI, 0.91–1.14; *p* = 0.732) [4].

Apart from sleep duration, other sleep factors, such as sleep disorders and trouble sleeping, contribute to an increased risk of hypertension [28,29]. Despite the major role of the above sleep problems, the importance of overall sleep quality cannot be ignored. A meta-analysis showed that poor sleep quality was significantly associated with a greater likelihood of hypertension (odds ratio, 1.48; *p* = 0.01) [29]. Additionally, a combined assessment of sleep behaviors suggested that the co-occurrence of insomnia and short sleep duration is associated with an increased risk of hypertension [30,31]. Consistent with this, we identified an association between overall sleep quality (defined as the sleep pattern) and an increased risk of hypertension. Compared with healthy sleep patterns, participants with poor sleep patterns had a higher risk of hypertension (OR = 1.90, 95% CI: 1.62 to 2.24, *p* < 0.001).

Several biological mechanisms have been proposed that relate sleep to the risk of hypertension. Sleep disorders can affect central nervous system regulation [32], hemodynamics [33], ventilation function [34] and biological rhythms [35,36], leading to physiological function alterations and pathological changes in blood pressure. In addition, 16 cross-sectional studies have suggested that a short sleep duration, higher total energy intake and higher total fat intake are related to hypertension [37]. One clinical study found that a short sleep duration was associated with decreased leptin levels, elevated ghrelin levels, and increased hunger and appetite [38]. There is also evidence that short sleep can promote one’s appetite for salt and suppress the excretion of salt in the renal fluid [39].

A major strength of this study is the large sample size from NHANES. Additionally, the complex nature of overall sleep was taken into consideration by integrating sleep quality and quantity (sleep duration, self-reported trouble sleeping and sleep disorder) into a single sleep measurement. There are some limitations to this study. First, owing to the cross-sectional nature of the study, the results could not be used to define a causal link between sleep and hypertension. Second, the sleep-related data were collected from self-reports instead of objective measurements; thus, there is a possibly of memory bias. The data collected were limited regarding sleep duration and referred only to weekdays; information on shift work and sleep duration from weekend was not provided. Night shifts and early morning shifts may cause acute sleep loss, and some studies support links between shift work, elevated blood pressure, and hypertension [40]. A recent study suggests that sleep debt, defined as a difference between weekday and weekend sleep deprivation of at least 2 h, was associated with poorer cardiovascular health in older females [41]. The forms of sleep disorders are many and varied, such as obstructive sleep apnea, insomnia, and delayed sleep phase disorder. Self-reported sleep disorders were not specifically classified, and the relationship between certain types of sleep disorders and hypertension could not be measured. Third, additional connections between sleep and hypertension were not considered. Although we controlled for a number of potential confounders, including demographics, health behaviors and clinical status, residual confounding factors might still be present. Furthermore, the status of prescription medication use was not included in the analysis, as there were many missing values. Glucocorticoids can disrupt sleep and cause hypertension, and this was not accounted for in our study [42].

## 5. Conclusions

In summary, the findings indicate that poor sleep patterns are associated with a higher risk of hypertension. Sleep, as a modifiable behavior, may be relevant for the primary prevention of hypertension. Further prospective studies are necessary to elucidate the causal link between sleep and hypertension.

## Figures and Tables

**Figure 1 ijerph-18-07867-f001:**
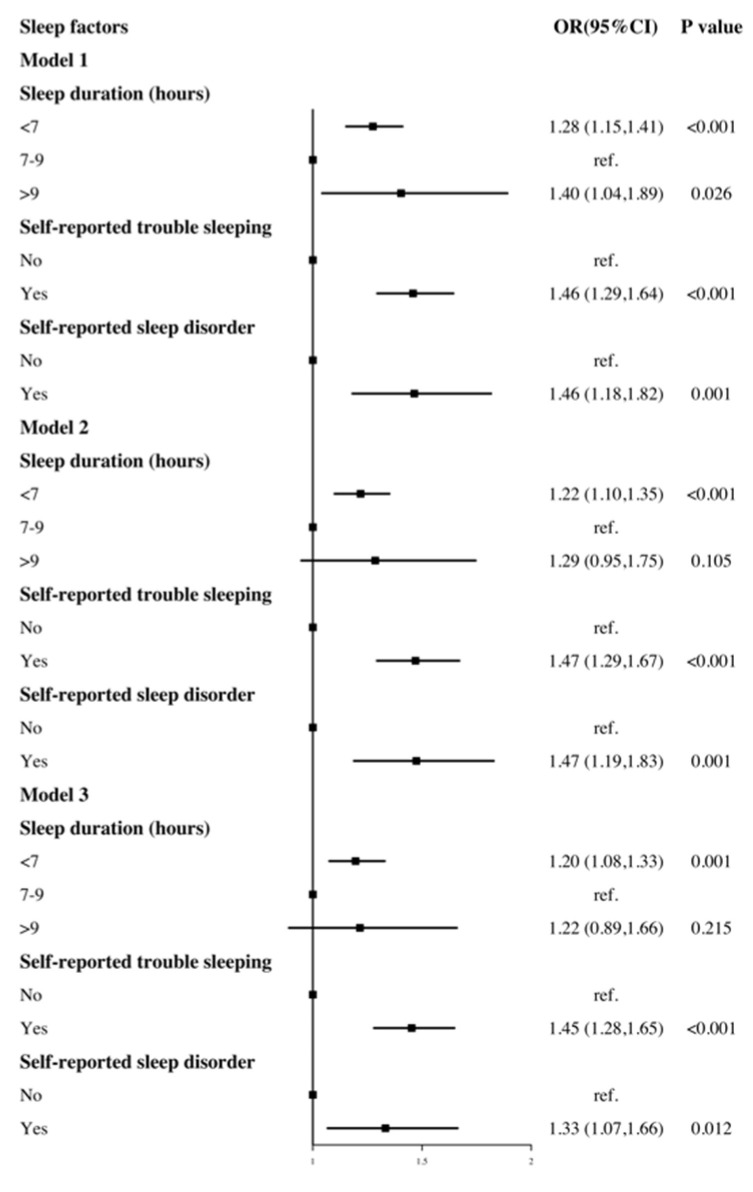
The association between sleep duration, diagnosed trouble sleeping diagnosed sleep disorder and hypertension. Model 1 adjusted for gender, age. Model 2 adjusted for gender, age, race, marital status, education level, health insurance. Model 3 adjusted for gender, age, race, marital status, education level, health insurance, BMI, diabetes, physical activity, sedentary time, smoke status, alcohol intake, DASH index.

**Figure 2 ijerph-18-07867-f002:**
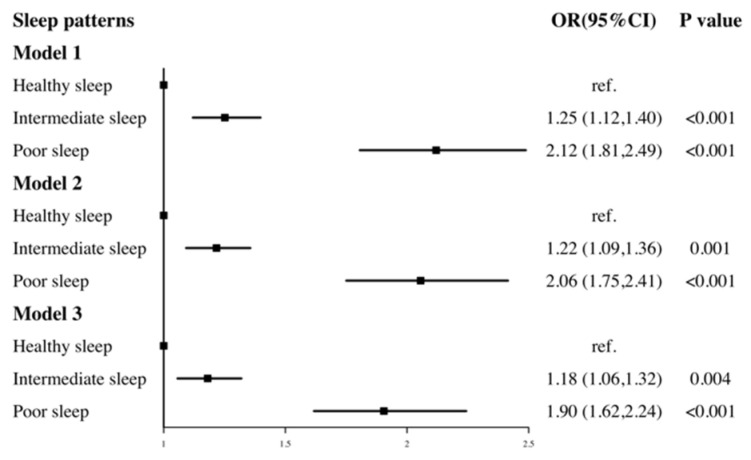
The association between sleep patterns and hypertension. Model 1 adjusted for gender, age. Model 2 adjusted for gender, age, race, marital status, education level, health insurance. Model 3 adjusted for gender, age, race, marital status, education level, health insurance, BMI, diabetes, physical activity, sedentary time, smoke status, alcohol intake, DASH index.

**Figure 3 ijerph-18-07867-f003:**
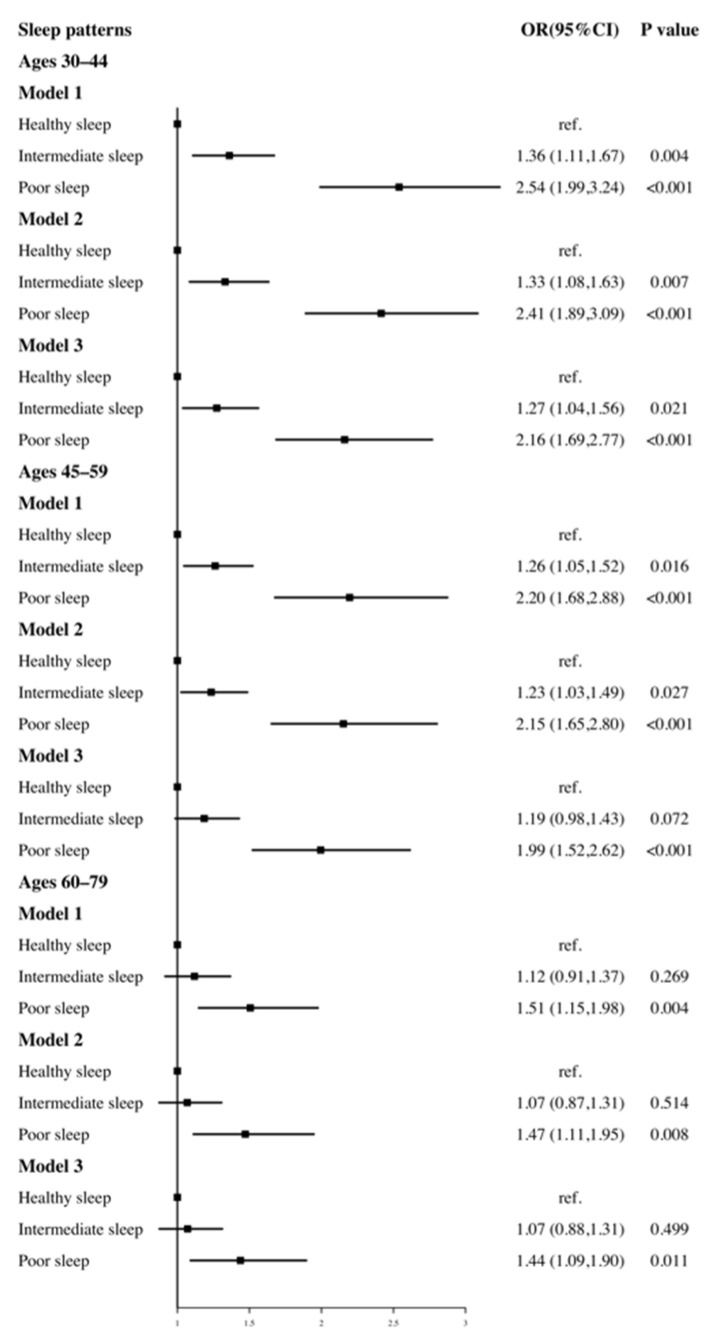
The association of sleep patterns with risk of hypertension after age stratification. Model 1 adjusted for gender, age. Model 2 adjusted for gender, age, race, marital status, education level, health insurance. Model 3 adjusted for gender, age, race, marital status, education level, health insurance, BMI, diabetes, physical activity, sedentary time, smoke status, alcohol intake, DASH index.

**Table 1 ijerph-18-07867-t001:** Study sample characteristics by sleep pattern status, the NHANES 2007–2014.

Characteristics	Sleep Patterns (Col %)			
	Healthy (*n* = 5675)	Intermediate (*n* = 4539)	Poor (*n* = 1952)	*p* Value
Demographics				
Age group (years)	49.8 ± 13.6	49.5 ± 13.0	51.1 ± 12.6	<0.001
30–44	2404 (42.4)	1853 (40.8)	663 (34.0)	
45–59	1675 (29.5)	1519 (33.5)	749 (38.4)	
60–79	1596 (28.1)	1167 (25.7)	540 (27.7)	
Gender				<0.001
Male	2743 (48.3)	2217 (48.8)	796 (40.8)	
Female	2932 (51.7)	2322 (51.2)	1156 (59.2)	
Race				<0.001
Non-white ^a^	3117 (54.9)	2659 (58.6)	928 (47.5)	
White	2558 (45.1)	1880 (41.4)	1024 (52.5)	
Marriage status				<0.001
Married/Living with partner	3992 (70.3)	2932 (64.6)	1105 (56.6)	
Widowed/Divorced/ Separated/Never married	1683 (29.7)	1607 (35.4)	847 (43.4)	
Education level				0.543
≤High school	2588 (45.6)	2119 (46.7)	895 (45.9)	
>High school	3087 (54.4)	2420 (53.3)	1057 (54.1)	
Health insurance				<0.001
No	1518 (26.7)	1093 (24.1)	353 (18.1)	
Yes	4157 (73.3)	3446 (75.9)	1599 (81.9)	
Clinical status				
BMI (kg/m^2^)				<0.001
<25	1745 (30.7)	1283 (28.3)	446 (22.8)	
≥25	3930 (69.3)	3256 (71.7)	1506 (77.2)	
Health behaviors				
DASH index				0.007
0–3.5	4270 (75.2)	3533 (77.9)	1507 (77.2)	
4–7.5	1405 (24.8)	1006 (22.2)	445 (22.8)	
Smoking status ^b^				<0.001
No	3388 (59.7)	2427 (53.5)	929 (47.6)	
Yes	2287 (40.3)	2112 (46.5)	1023 (52.4)	
Alcohol intake (gm/day)				0.015
0–24.9	4815 (84.8)	3839 (84.6)	1703 (87.2)	
25.1–550.1	860 (15.2)	700 (15.4)	249 (12.8)	
Physical activity (met-h/week)				<0.001
0	1279 (22.5)	1077 (23.7)	547 (28.0)	
>0	4396 (77.5)	3462 (76.3)	1405 (72.0)	
Sedentary time (hours)				<0.001
<5	2611 (46.0)	2023 (44.6)	765 (39.2)	
≥5	3064 (54.0)	2516 (55.4)	1187 (60.8)	

Abbreviations: *n*: sample size; BMI, body mass index (calculated as weight in kilograms divided by height in square meters); DASH, Dietary Approaches to Stop Hypertension [13]; ^a^ Non-white: Mexican American, other Hispanic, non-Hispanic Black, other race—including multi-racial; ^b^ Smoke 100 cigarettes (or other tobacco) in entire life.

**Table 2 ijerph-18-07867-t002:** Characteristics of participants according to hypertension status, the NHANES 2007–2014.

Characteristics	Hypertension (Col %)		
	No (*n* = 7571)	Yes (*n* = 4595)	*p* Value
Demographics			
Age (years)	46.1 ± 11.9	56.2 ± 13.0	<0.001
30–44	3911 (79.5)	1009 (20.5)	
45–59	2436 (61.8)	1507 (38.2)	
60–79	1224 (37.1)	2079 (62.9)	
Gender			0.261
Male	3552 (61.7)	2204 (38.3)	
Female	4019 (62.7)	2391 (37.3)	
Race			0.189
Non-white ^a^	4137 (61.7)	2567 (38.3)	
White	3434 (62.9)	2028 (37.1)	
Marital status			<0.001
Married/Living with partner	5198 (64.8)	2831 (35.3)	
Widowed/Divorced/Separated/ Never married	2373 (57.4)	1764 (42.6)	
Education level			<0.001
≤High school	3300 (58.9)	2302 (41.1)	
>High school	4271 (65.1)	2293 (34.9)	
Health insurance			<0.001
No	2080 (70.2)	884 (29.8)	
Yes	5491 (59.7)	3711 (40.3)	
Clinical status			
BMI (kg/m^2^)			<0.001
<25	2544 (73.2)	930 (26.8)	
≥25	5027 (57.8)	3665 (42.2)	
Health behaviors			
Sleep duration (hours)			<0.001
<7	4546 (64.4)	2514 (35.6)	
7–9	2888 (59.2)	1987 (40.8)	
>9	137 (59.3)	94 (40.7)	
Self-reported trouble sleeping			<0.001
No	6043 (65.2)	3227 (34.8)	
Yes	1528 (52.8)	1368 (47.2)	
Self-reported sleep disorder			<0.001
No	7145 (63.3)	4142 (36.7)	
Yes	426 (48.5)	453 (51.5)	
Sleep patterns			<0.001
Healthy sleep	3757 (66.2)	1918 (33.8)	
Intermediate sleep	2852 (62.8)	1687 (37.2)	
Poor sleep	962 (49.3)	990 (50.7)	
DASH index			0.637
0–3.5	5783 (62.1)	3527 (37.9)	
4–7.5	1788 (62.6)	1068 (37.4)	
Smoking status ^b^			<0.001
No	4350 (64.5)	2394 (35.5)	
Yes	3221 (59.4)	2201 (40.6)	
Alcohol intake (gm/day)			0.378
0–24.9	6462 (62.4)	3895 (37.6)	
25.1–550.1	1109 (61.3)	700 (38.7)	
Physical activity (met-h/week)			<0.001
0	1595 (54.9)	1308 (45.1)	
>0	5976 (64.5)	3287 (35.5)	
Sedentary time (hours)			<0.001
<5	3459 (64.1)	1940 (35.9)	
≥5	4112 (60.8)	2655 (39.2)	

Abbreviations: *n*: sample size; BMI, body mass index (calculated as weight in kilograms divided by height in square meters); DASH, Dietary Approaches to Stop Hypertension [13]; ^a^ Non-white: Mexican American, other Hispanic, non-Hispanic Black, other race—including multi-racial; ^b^ Smoke 100 cigarettes (or other tobacco) in entire life.

## Data Availability

The datasets used and analyzed during the current study are available from the corresponding author on reasonable use. The NHANES public database is available at https://www.cdc.gov/nchs/nhanes/, accessed on 1 May 2021.

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
