# Peer review of "Relationship between Sleep and Hypertension: Findings from the NHANES (2007–2014)"

_ijerph, 2021, doi:10.3390/ijerph18157867_

Round 1
Reviewer 1 Report
Manuscript entitled “Relationship between Sleep and Hypertension: Findings from the NHANES (2007–2014)” reports on association between sleep length and disorders.
Introduction should provide references to some length and norms of healthy sleep and general average in the population (see: doi: 10.5664/jcsm.7036, doi: 10.2147/NSS.S163071)
Covariates and statistical analysis include repetition which causes disappearance of a proper flow in the manuscript. This should be rewritten.
The inclusion of individuals with diabetes seems to be an unnecessary confounding factor as complications are vast and hart to accent for. If authors want to keep it in the analysis it should be widely discussed in the discussion.
Limitations of the study should be greatly expanded and highlighted, as the data collected is very limited regarding sleep and refers only to the weekdays. It does not provide information on shift work, on weekend – those factors greatly influence quality of sleep being a risk factor for complications even with proper number of hours of sleep in the week. Furthermore, additional connections between sleep and hypertension are not considered, one of them hyperthyroidism, which shortens time of sleep and might cause hypertension. Other information that would be helpful is medication that was used by the participants – if it is not available consider it in the limitations as for example glucocorticosteroids can disrupt sleep and cause hypertension (see doi: 10.1016/j.smrv.2020.101380).
Abbreviations should be only included in the text, if they are further used. While abbreviations should not be used without expansions. This happens throughout the whole manuscript.
Tables should be self-explanatory and should have descriptions of abbreviations used within them.
Tense in manuscript sometimes changes from past to present.
Author Response
Response to Reviewer Comments
Reviewer 1
Comments and Suggestions for Authors
Manuscript entitled “Relationship between Sleep and Hypertension: Findings from the NHANES (2007–2014)” reports on association between sleep length and disorders.
Point 1: Introduction should provide references to some length and norms of healthy sleep and general average in the population (see: doi: 10.5664/jcsm.7036, doi: 10.2147/NSS.S163071)
Response 1:
Owing to high interindividual and intraindividual variations in sleep at different ages across the lifespan [2], the American National Sleep Foundation has recommended a 7–9 h sleep duration for adults [3].
Reference:
[2] Hertenstein E, Gabryelska A, Spiegelhalder K, Nissen C, Johann AF, Umarova R, et al. Reference Data for Polysomnography-Measured and Subjective Sleep in Healthy Adults. J Clin Sleep Med. 2018;14(4):523-32
[3] Chaput JP, Dutil C, Sampasa-Kanyinga H. Sleeping hours: what is the ideal number and how does age impact this? Nat Sci Sleep. 2018;10:421-30.
Point 2. Covariates and statistical analysis include repetition which causes disappearance of a proper flow in the manuscript. This should be rewritten.
Response 2:
The baseline characteristics of the participants were categorized by sleep patterns and hypertension status.
We applied the survey weight from the mobile examination center (MEC exam weight) to all analyses to account for the cluster sample design, oversampling, poststratification, survey nonresponse and sampling frame, as suggested. The weight of the survey allows it to be extended to the civilian noninstitutionalized US population [22,23]. Weighted logistic regression was used to calculate the odds ratios (ORs) and 95% confidence intervals (95% CIs) to assess the relationship between each sleep factor and hypertension. The same analyses were performed to explore the relationship between sleep patterns and hypertension.
Statistical analyses were performed with STATA version 14.0 (Stata Corp LP, College Station, Texas, USA). The forest graphs were plotted using R version 3.5.3. A p-value < 0.05 was defined as statistically significant.
Reference:
[22] National Health and Nutrition Examination Survey: Estimation Procedures, 2007-2010 https://www.cdc.gov/nchs/data/series/sr_02/sr02_159.pdf
[23] National Health and Nutrition Examination Survey: Estimation Procedures, 2011–2014 https://www.cdc.gov/nchs/data/series/sr_02/sr02_177.pdf
Point 3: The inclusion of individuals with diabetes seems to be an unnecessary confounding factor as complications are vast and hart to accent for. If authors want to keep it in the analysis it should be widely discussed in the discussion.
Response 3:
Participants with diabetes (n=1617) or borderline (n=323) were excluded and 12166 adults were included.
Point 4: Limitations of the study should be greatly expanded and highlighted, as the data collected is very limited regarding sleep and refers only to the weekdays. It does not provide information on shift work, on weekend – those factors greatly influence quality of sleep being a risk factor for complications even with proper number of hours of sleep in the week. Furthermore, additional connections between sleep and hypertension are not considered, one of them hyperthyroidism, which shortens time of sleep and might cause hypertension. Other information that would be helpful is medication that was used by the participants – if it is not available consider it in the limitations as for example glucocorticosteroids can disrupt sleep and cause hypertension (see doi: 10.1016/j.smrv.2020.101380).
Response 4:
There are some limitations to this study. First, owing to the cross-sectional nature of the study, the results could not be used to define a causal link between sleep and hypertension. Second, the sleep-related data were collected from self-reports instead of objective measurements; thus, there is a possibly of memory bias. The data collected were limited regarding sleep duration and referred only to weekdays; information on shift work and sleep duration from weekend was not provided. Night shifts and early morning shifts may cause acute sleep loss, and some studies support links between shift work, elevated blood pressure, and hypertension [41]. A recent study suggests that sleep debt, defined as a difference between weekday and weekend sleep deprivation of at least 2 hours, was associated with poorer cardiovascular health in older females [42]. The forms of sleep disorders are many and varied, such as obstructive sleep apnea, insomnia, and delayed sleep phase disorder. Self-reported sleep disorders were not specifically classified, and the relationship between certain types of sleep disorders and hypertension could not be measured. Third, additional connections between sleep and hypertension were not considered. Although we controlled for a number of potential confounders, including demographics, health behaviors and clinical status, residual confounding factors might still be present. Furthermore, the status of prescription medication use was not included in the analysis, as there were many missing values. Glucocorticoids can disrupt sleep and cause hypertension, and this was not accounted for in our study [43].
Reference:
[41] Kecklund G, Axelsson J. Health consequences of shift work and insufficient sleep. BMJ. 2016;355:i5210.
[42] Cabeza de Baca T, Chayama KL, Redline S, Slopen N, Matsushita F, Prather AA, et al. Sleep debt: the impact of weekday sleep deprivation on cardiovascular health in older women. Sleep. 2019;42(10).
[43] Szmyd B, Rogut M, Bialasiewicz P, Gabryelska A. The impact of glucocorticoids and statins on sleep quality. Sleep Med Rev. 2021;55:101380.
Point 5: Abbreviations should be only included in the text, if they are further used. While abbreviations should not be used without expansions. This happens throughout the whole manuscript.
Response 5:
Thank you for your advice, the English language editing has been performed as suggested.
Point 6: Tables should be self-explanatory and should have descriptions of abbreviations used within them.
Response 6:
Thank you for your advice, the English language editing has been performed as suggested.
Point 7: Tense in manuscript sometimes changes from past to present.
Response 7:
Thank you for your advice, the English language editing has been performed as suggested.
Reviewer 2
Comments and Suggestions for Authors
Overview:
The paper is a reanalysis of the NHANES study, investigating the relationship between reported sleep problems and hypertension.
There are a number of issues that need to be addressed:
General/Larger Issues:
Point 1: It is difficult to understand some of the subtler concepts in the paper due primarily to issues around grammatical and other language issues. – recommend English language editing – a good example is lines 145-151 where a critical point is made but the wording is confusing and ambiguous.
The authors should state what they measured when interpreting the results. Instead of saying people with a sleep disorder, they should always state that they are people who report a sleep disorder, there is a difference, particularly when it come to sleep where by definition they cannot directly observe the events. This would still allow for interesting results. Alternatively it should be demonstrated that people who report short sleep for instance actually have short sleep to some level of confidence and so on for the other variables.
Response 1:
Thank you for your advice, the English language editing has been performed as suggested.
According to the suggestion, we have replaced “diagnosed sleep xxx” with “self-reported sleep xxx”
For lines 145-151:
Figure 1 depicts the association between each sleep factor and hypertension, including the adjustments. In the age- and gender-adjusted model, a short sleep duration (OR = 1.28, 95% CI: 1.15 to 1.41, p <0.001), long sleep duration (OR = 1.40, 95% CI: 1.04 to 1.89, p =0.026), self-reported trouble sleeping (OR = 1.46, 95% CI: 1.29 to 1.64, p <0.001) and sleep disorder (OR = 1.46, 95% CI: 1.18 to 1.82, p =0.001) were all associated with the risk of hypertension. After additionally adjusting for race, marital status, education level, health insurance, BMI, physical activity, sedentary time, smoking status, alcohol intake, and the DASH index (fully adjusted models), these associations remained statistically significant except for a long sleep duration (OR =1.22, 95% CI: 0.89 to 1.66, p =0.215).
Point 2: The authors should address the very large gender difference between reported male and female sleep problems which is much larger than would be expected and potentially in the wrong direction.
Response 2:
Men and women have different complaints on the same sleep questionnaire, which may have an impact on these gender-specific associations (DOI: 10.1093/sleep/27.2.305). And hormonal influences may play an important role in the relationship between sleep and hypertension, especially during the premenopausal period (DOI: 10.1097/HJH.0b013e328335d076). In the case of a large number of missing values on menstrual status, we should be cautious about gender stratification, so we decided to delete this part to avoid misleading.
Point 3: The authors need to demonstrate why their finding are novel as it seems that these findings are just weaker confirmations of existing studies
Response 3:
A major strength of this study is the large sample size from NHANES. Additionally, the complex nature of overall sleep was taken into consideration by integrating sleep quality and quantity (sleep duration, self-reported trouble sleeping and sleep disorder) into a single sleep measurement.
Point 4. Minor Language Issues:
Examples of odd phrasing that make things difficult to understand are on:
4.1 Line 10 – missing ‘a’ before standardized
Response 4.1:
Thank you for your advice. It has been corrected in the manuscript according to the suggestion.
4.2 Line 12
Response 4.2:
We included three sleep factors (sleep duration, self-reported trouble sleeping and sleep disorder) to generate an overall sleep score, ranging from 0 to 3. We then defined the sleep pattern as ‘healthy sleep pattern’ (overall sleep score =3), ‘intermediate sleep pattern’ (overall sleep score =2), and ‘poor sleep pattern’ (0≤ overall sleep score ≤ 1) based on the overall sleep score.
4.3 Line 26 – ‘is’ instead of ‘are’
Response 4.3:
Thank you for your advice. It has been corrected in the manuscript according to the suggestion.
4.4 Line 37 – no need for ‘the’
Response 4.4:
Thank you for your advice. It has been corrected in the manuscript according to the suggestion.
4.5 Line 42-44 - ??
Response 4.5:
Sleep behaviors are multifaceted, and individual sleep behaviors are usually related to each other in a compensatory way [13,14]. In most previous studies, sleep behaviors were considered individually, without considering the complexity and correlation between sleep parameters [13]. Limited studies have been performed to assess sleep behaviors jointly; these have suggested that sleep duration in combination with other unhealthy sleep behaviors, such as trouble sleeping and sleep disorders, is associated with a higher risk of CVD and hypertension [15-17].
Reference:
[13] Fan M, Sun D, Zhou T, Heianza Y, Lv J, Li L, et al. Sleep patterns, genetic susceptibility, and incident cardiovascular disease: a prospective study of 385 292 UK biobank participants. Eur Heart J. 2020;41(11):1182-9.
[14] Li X, Xue Q, Wang M, Zhou T, Ma H, Heianza Y, et al. Adherence to a Healthy Sleep Pattern and Incident Heart Failure: A Prospective Study of 408 802 UK Biobank Participants. Circulation. 2021;143(1):97-9.
[15] Fernandez-Mendoza J, Vgontzas AN, Liao D, Shaffer ML, Vela-Bueno, A., Basta M; et al. Insomnia with objective short sleep duration and incident hypertension: The Penn State Cohort. Hypertension. 2012, 60, 929–935.
[16] St-Onge MP, Grandner MA, Brown D, Conroy MB, Jean-Louis, G., Coons M; et al. Sleep Duration and Quality: Impact on Lifestyle Behaviors and Cardiometabolic Health: A Scientific Statement from the American Heart Association. Circulation. 2016, 134, e367–e86.
[17] Bathgate CJ, Fernandez-Mendoza, J.; Insomnia, Short Sleep Duration, and High Blood Pressure: Recent Evidence and Future Directions for the Prevention and Management of Hypertension. Curr Hypertens Rep. 2018, 20, 52.
4.6 Line 74-75 – says that people responded ‘yes’ to a question that required a numerical response
Response 4.6:
Nighttime sleep hours were obtained by the response to “How much sleep do you usually get at night on weekdays or workdays?”. The sleep duration was categorized as short (<7 h per night), normal (7-9 h per night), or long (>9 h per night), and 7-9 h per night was used as the reference group.
4.7 Line 108 – tense
Response 4.7:
Thank you for your advice. It has been corrected in the manuscript according to the suggestion.
Point 5: Some of the language is unscientific:
5.1 Line 77 – cant say ‘confirmed’ – this is not appropriate even if true which it is not
Response 5.1:
The response to “Have you ever told a doctor or other health professional that you have trouble sleeping?” was used to assess trouble sleeping.
5.2 Line 104 – back up ‘reliable’ claim
Response 5.2:
Dietary intake was determined using valid 24-hour dietary recalls, as detailed in a previous study [19].
Reference:
[19] Rehm CD, Penalvo JL, Afshin A, Mozaffarian D. Dietary Intake Among US Adults, 1999-2012. JAMA. 2016;315(23):2542-53.
Point 6: Other Minor Issues
6.1 The manuscript would also be more readable if many of the unnecessary acronyms (3 or more in a sentence, things abbreviated but never or rarely used again) were removed and their usage made consistent.
Response 6.1:
Thank you for your advice. It has been corrected in the manuscript according to the suggestion.
6.2 Figure 2 not discussed, what are the implications for the models
Response 6.2:
Apart from sleep duration, other sleep factors, such as sleep disorders and trouble sleeping, contribute to an increased risk of hypertension [29,30]. Despite the major role of the above sleep problems, the importance of overall sleep quality cannot be ignored. A meta-analysis showed that poor sleep quality was significantly associated with a greater likelihood of hypertension (odds ratio, 1.48; P = 0.01) [31]. Additionally, a combined assessment of sleep behaviors suggested that the co-occurrence of insomnia and short sleep duration is associated with an increased risk of hypertension [30,32]. Consistent with this, we identified an association between overall sleep quality (defined as the sleep pattern) and an increased risk of hypertension. Compared with healthy sleep patterns, participants with poor sleep patterns had a higher risk of hypertension (OR = 1.90, 95% CI: 1.62 to 2.24, p <0.001).
Reference:
[29] Van Ryswyk E, Mukherjee S, Chai-Coetzer CL, Vakulin, A., McEvoy RD. Sleep Disorders, Including Sleep Apnea and Hypertension. Am J Hypertens. 2018, 31, 857–864.
[30] Thomas SJ, Calhoun, D. Sleep, insomnia, and hypertension: Current findings and future directions. J Am Soc Hypertens. 2017, 11, 122–129.
[31] Lo K, Woo B, Wong M, Tam, W. Subjective sleep quality, blood pressure, and hypertension: A meta-analysis. J Clin Hypertens (Greenwich). 2018, 20, 592–605.
[32] Meng, L., Zheng, Y., Hui, R. The relationship of sleep duration and insomnia to risk of hypertension incidence: A meta-analysis of prospective cohort studies. Hypertens Res. 2013, 36, 985–995.

Reviewer 2 Report
Overview:
The paper is a reanalysis of the NHANES study, investigating the relationship between reported sleep problems and hypertension.
There are a number of issues that need to be addressed:
General/Larger Issues:
- It is difficult to understand some of the subtler concepts in the paper due primarily to issues around grammatical and other language issues. – recommend English language editing – a good example is lines 145-151 where a critical point is made but the wording is confusing and ambiguous.
- The authors should state what they measured when interpreting the results. Instead of saying people with a sleep disorder, they should always state that they are people who report a sleep disorder, there is a difference, particularly when it come to sleep where by definition they cannot directly observe the events. This would still allow for interesting results. Alternatively it should be demonstrated that people who report short sleep for instance actually have short sleep to some level of confidence and so on for the other variables.
- The authors should address the very large gender difference between reported male and female sleep problems which is much larger than would be expected and potentially in the wrong direction
- The authors need to demonstrate why their finding are novel as it seems that these findings are just weaker confirmations of existing studies
Minor Language Issues:
- Examples of odd phrasing that make things difficult to understand are on:
- Line 10 – missing ‘a’ before standardized
- Line 12
- Line 26 – ‘is’ instead of ‘are’
- Line 37 – no need for ‘the’
- Line 42-44 - ??
- Line 74-75 – says that people responded ‘yes’ to a question that required a numerical response
- Line 108 – tense
- Some of the language is unscientific :
- Line 77 – cant say ‘confirmed’ – this is not appropriate even if true which it is not
- Line 104 – back up ‘reliable’ claim
Other Minor Issues
- The manuscript would also be more readable if many of the unnecessary acronyms (3 or more in a sentence, things abbreviated but never or rarely used again) were removed and their usage made consistent.
- Figure 2 not discussed, what are the implications for the models
Author Response

(The authors gave the same response as above.)

Round 2
Reviewer 1 Report
Authors address all the comments and greatly improved the manuscript.
Reviewer 2 Report
all concerns addressed